# Factors Related to the COVID-19 Prevention Behaviors of Adolescents: Focusing on Six Middle and High Schools in Korea

**DOI:** 10.3390/healthcare11202779

**Published:** 2023-10-20

**Authors:** Shinae Lee, Gye-Hyun Jung, Hye-Young Song

**Affiliations:** 1Department of Nursing, Daegu Health College, Daegu 41453, Republic of Korea; shinaelee@dhc.ac.kr; 2Department of Nursing, Jeonbuk Science College, Jeonbuk 56204, Republic of Korea; ghjung@jbsc.ac.kr; 3College of Nursing, Woosuk University, 443 Samnye-ro, Wanju-gun 55338, Republic of Korea

**Keywords:** adolescent, COVID-19, prevention behavior

## Abstract

The COVID-19 pandemic has been one of the worst infectious disasters in human history. The best way to minimize COVID-19 transmission is to follow preventive measures. This study aimed to examine the factors influencing adolescents’ COVID-19 prevention behaviors. The study was conducted online from 1 to 15 February 2023 with 196 adolescents aged between 13 and 18 years of age. The collected data were analyzed using descriptive statistics, Pearson’s correlation, the independent t-test, analysis of variance (ANOVA), and multiple hierarchical regression analysis. Adolescents’ COVID-19 prevention behaviors were influenced by intrapersonal factors, such as knowledge of and attitudes toward COVID-19, and interpersonal factors, such as social support. Community and governmental factors had no impact. Public health education strategies should be planned to include friends and family members in programs for preventing new infectious diseases such as COVID-19 so that adolescents can learn and share what they have learned, correct wrong behavior, and understand and change infection prevention behavior. In addition, it is necessary to actively support the development of public health education with appropriate contents in accordance with the characteristics and preferences of adolescents.

## 1. Introduction

COVID-19, an infectious disease that emerged in December 2019, has caused significant harm. In March 2020, the World Health Organization (WHO) designated COVID-19 as a pandemic and issued guidelines to prevent its spread [1,2]. Many nations have enacted robust mitigation measures, such as lockdowns and social distancing, to curb viral transmission [1,3,4]. Given COVID-19 transmission occurs via droplets and physical contact, people have been asked to strictly maintain personal hygiene practices, including wearing masks, hand hygiene, and social distancing [3,5,6]. Notably, South Korea garnered praise for its COVID-19 response within the Organization for Economic Cooperation and Development (OECD) group of 33 countries [7]. Nonetheless, by February 2023, the cumulative number of confirmed cases in South Korea had reached 30,297,315. Analysis of the age distribution in these cases revealed that the highest proportion (15.2%) of individuals were in their 40s, while adolescents constituted 12.4% of the total [5]. Sustaining preventive measures, such as hand washing and physical distancing, might be somewhat more challenging compared to the initial stages of the outbreak [8]. Adherence to COVID-19 preventive practices remains paramount and serves as the primary strategy against infection. Recognizing the potential wane in persistent adherence to these preventive behaviors, it is imperative to comprehend the factors influencing such adherence, aiding in the targeted development of interventions to promote these behaviors [9].

Cross-sectional studies have been conducted in many countries to investigate the effectiveness of preventive interventions in reducing COVID-19 infection and severity. Despite wearing masks and being vaccinated, COVID-19 infection rates differ depending on age [10]. Accordingly, research on preventive behaviors according to age is necessary, and adolescents are more vulnerable to COVID-19 than adults [11]. They are mostly gathered in confined spaces at schools, and most classrooms in schools are closed and poorly ventilated; one student’s infection can spread to the entire school and not just the members of the classroom. Furthermore, it can spread to the students’ families and social contacts [12]. In addition, owing to the pandemic, adolescents worldwide have experienced challenges such as changes in family dynamics, death of family members, loss of friendships due to social isolation, and learning delays [4,13,14]. This indicates that COVID-19 has had more severe effects on adolescents than on adults, making prevention and response to COVID-19 even more important [15,16].

In general, the most common symptoms of COVID-19 are fatigue, sleep disorders, headaches, and respiratory symptoms [16]. However, adolescents show less severe COVID-19 symptoms than adults or even no symptoms at all [11], which has led to a low awareness of infectious diseases among them and resulted in higher rates of infection compared to other age groups [17]. There is increasing evidence that asymptomatic patients can spread the virus to others [18]. Furthermore, the long-COVID infection rate among children and adolescents aged 0 to 18 is 25.24%, a severe level, and the risk increases with age. Thus, infection prevention is even more important for adolescents [15].

In particular, high school students in South Korea are more likely to be exposed to health-risk behaviors, such as stress, lack of exercise, sleep deprivation, and unstable eating habits owing to entrance examination-driven classes and after-school learning activities [15]. During outbreaks, children and adolescents are more susceptible to infectious disease transmission than other age groups. An outbreak of 37 confirmed cases in a high school in Seoul is a representative example of such susceptibility [19]. Thus, it is essential to develop strategies to raise awareness of the importance of COVID-19 infection prevention behaviors among adolescents, especially middle and high school students, to help them practice those behaviors [17,18].

As the importance of COVID-19 prevention behaviors is recognized worldwide, many studies have been conducted on adolescents. However, the majority of research results, which primarily focus on the researchers respective countries rather than Korea, may not directly translate to the context of Korean adolescents due to different cultural and economic factors. In Korea, research has been conducted on college students and adults [20,21], but there are almost no studies conducted on middle school and high school students [12]. Research on infectious disease prevention behaviors in South Korea has mainly focused on individual variables from the Health Belief Model [22,23] and the Theory of Reasoned Action [24]. Infection prevention behaviors are determined not only by individual characteristics but also by relationships with family and neighbors [25] and the physical environment [23]. Therefore, it is necessary to understand the social and physical environments surrounding individual and community context [23,26,27].

Ecological models are suitable for understanding the complex interactions between individuals and their environments [28,29]. This theory defines the interactions between humans and the environment as an ecological system. It has the advantage of allowing multidimensional and diverse interventions by considering the interactions between different factors that influence individual behaviors [28,30]. Thus, the ecological model can be helpful in adopting a multifaceted approach to understand the factors that influence Korean adolescents’ COVID-19 prevention behaviors. We considered the following factors: intrapersonal factors (knowledge of COVID-19 and attitudes toward COVID-19), interpersonal factors (social support), and community factors (community and government responses to COVID-19), as shown in Figure 1.

Typically, awareness of the community response to COVID-19 has been managed by recommending that community members cancel or refrain from group gatherings or strictly follow precautions according to social distancing steps. Meanwhile, awareness of the government’s response to COVID-19 in Korea has been consistently promoted by informing the public of the current situation regarding national disaster safety and sending out disaster text alerts on preventive measures and precautions [31]. Given the possibility of emerging infectious diseases, active government and community responses are effective and important ways to protect individuals from diseases and prevent community outbreaks. Indeed, they have been an important factor in practicing COVID-19 prevention behaviors [21]. Therefore, the purpose of this study was to identify the factors influencing adolescents’ COVID-19 prevention behaviors by including awareness factors of community and government responses that have not been explored in previous studies and to provide the basic data necessary for the development of specific recommendations and health education programs for COVID-19 prevention behaviors among adolescents.

## 2. Materials and Methods

### 2.1. Study Design and Participants

This study employed a cross-sectional design. The participants in this study were convenience samples from three middle schools and three high schools in S City, J City, and D City in South Korea. Note that the regional variance in the attitudes and behaviors of adolescents is not significant because the school system in Korea remains relatively consistent across regions over the country. In order to account for even minor variances, we have taken steps to ensure sample diversity by collecting data from three major cities located in geometrically distributed regions of Korea, including a large metropolitan area.

The inclusion criteria were as follows: adolescents between the ages of 13 and 18 years of age, with parental consent, who could understand the questionnaire. The sample size was calculated based on Cohen’s sampling rationale [32], with a medium effect size for regression analysis, a significance level of 0.05, and a power of 0.95. The minimum sample size, calculated using the G*Power 3.1 software [33], was 189. Of the 200 questionnaires initially collected, 196 were used in the final analysis after excluding four questionnaires with insufficient responses.

### 2.2. Data Collection

Data were collected between 1 and 15 February 2023. Recruitment was conducted using the school’s online bulletin board after receiving cooperation from a schoolteacher. The schoolteachers sent the research consent form and the URL of the research subject consent form to adolescents who expressed their intention to participate in the research. Participants completed an electronic questionnaire via mobile phone. The time required to complete the survey was approximately 15 min. Convenience store gift certificates were given as gifts after completing the survey, with the subjects’ consent.

### 2.3. Measures

#### 2.3.1. General Characteristics

In the questionnaire, we explored participants’ general characteristics, including gender, grade, residential regions, subjective physical health status, subjective mental health status, daily life limitations due to COVID-19, and whether they had been tested for COVID-19.

#### 2.3.2. Infection Prevention Behaviors

COVID-19-related infection prevention behaviors were measured using a tool developed according to the Korea Centers for Disease Control and Prevention’s COVID-19 Response Guidelines [21]. The evaluation tool for infection prevention behaviors for COVID-19 comprised 12 questions related to the practice of wearing a mask, handwashing, social distancing, cough etiquette, avoiding public transportation, and avoiding crowded places. Responses were rated on a five-point Likert scale, with higher scores indicating higher levels of practice in infection prevention behaviors. At the time of development, the tool had a Cronbach’s α of 0.76 [21]. We found a Cronbach’s α of 0.88.

#### 2.3.3. Knowledge about COVID-19

COVID-19 knowledge was measured using a tool developed based on the COVID-19 Response Guidelines [34]. The tool asked questions related to the knowledge of COVID-19, including symptoms of the disease, its transmission mode, prevention methods, and actions taken in cases of suspected symptoms. The questions were dichotomous, with a score of 1 for a correct answer and 0 for an incorrect answer, and the total score ranged from 0 to 16, with higher scores indicating greater knowledge of COVID-19. At the time of development, the tool had a Cronbach’s α of 0.72 [34]. We found a Cronbach’s α of 0.73.

#### 2.3.4. Attitudes toward COVID-19

Attitudes toward COVID-19 were measured using a tool modified by Baek [35] and developed by Park [36]. This included 12 questions on a five-point Likert scale, and questions 8 and 11, which were negative statements, were reverse-scaled. Higher scores indicated more positive attitudes toward COVID-19. At the time of development, the tool had a Cronbach’s α of 0.71 [35]. We found a Cronbach’s α of 0.80.

#### 2.3.5. Social Support

Social support was measured using the Multidimensional Scale of Perceived Social Support (MSPSS) developed by Zimet et al. [37] and translated by Shin and Lee [38]. The MSPSS comprises twelve questions answered on a five-point Likert scale, with four questions each on the support of family, friends, and significant others. Higher scores indicate higher social support. At the time of development, the tool had a Cronbach’s α of 0.85 [38]. We found a Cronbach’s α of 0.94.

#### 2.3.6. Community Responses to COVID-19

Questions on the community’s response to COVID-19 aimed to understand how participants perceived it. We used an adapted and supplemented version [21] of the COVID-19 Community Response Tool developed by Lee and You [39] that contained eight questions on a five-point Likert scale, with higher scores indicating more positive perceptions of the community’s response to COVID-19. At the time of development, the tool had a Cronbach’s α of 0.88 [39]. We found a Cronbach’s α of 0.92.

#### 2.3.7. Government Response to COVID-19

The questions on the government response to COVID-19 aimed to understand how the subjects perceived it. We used an adapted version [21] of the COVID-19 Government Response Tool developed by Lee and You [39]. The tool had five questions: two on treatment and quarantine and three on the provision of information. Responses were recorded on a five-point Likert scale, with higher scores indicating more positive perceptions of the government’s response to COVID-19. At the time of development, the tool had a Cronbach’s α of 0.88 [39]. We found a Cronbach’s α of 0.91.

### 2.4. Ethical Considerations

This study was approved by the Institutional Review Board of the investigators’ institution (IRB No. WS-2023-03). The purpose, methods, voluntary participation, free withdrawal of consent, and lack of penalty for not participating in the study were explained to the participants on a subject information sheet. The participants were asked about their voluntary participation in the study through an online consent form. No personally identifiable information, such as names or phone numbers, was collected, and the collected data were encrypted and stored on a locked personal computer. Participants were informed that the data collected through the survey would be used for research purposes only and would be destroyed after the study.

### 2.5. Analysis

The collected data were analyzed using the SPSS/WIN 24.0 program as follows. First, we conducted a frequency analysis to identify the participants’ general characteristics. Second, descriptive statistical analyses were performed to understand adolescents’ knowledge of COVID-19; attitudes toward COVID-19, social support, the government response to COVID-19, and the community response to COVID-19; and levels of infection prevention behaviors. Third, Pearson’s correlation analysis was conducted to test the correlation between the aforementioned factors and infection prevention behaviors. Fourth, an independent sample t-test and one-way ANOVA were conducted to investigate whether these factors and infection prevention behaviors differed significantly according to the adolescents’ general characteristics. Fifth, multiple hierarchical regression analysis was performed to evaluate the factors that affected adolescents’ infection prevention behaviors.

## 3. Results

### 3.1. General Characteristics of the Participants

The participants’ general characteristics are presented in Table 1. This study included 69 (35.2%) men and 127 (64.8%) women. 98 participants each were middle (50%) or high school (50%) students. Regarding residential area, 108 (55.1%) lived in the capital region and 88 (44.9%) lived in non-capital regions. Subjective physical health status was reported as “Below average” by 87 (44.4%) and “Good” by 109 (55.6%), while subjective mental health status was reported as “Below average” by 91 (46.4%) and “Good” by 105 (53.6%). Life limitations due to COVID-19 were reported as “None” by 111 (56.6%) and “Yes” by 85 (43.4%). COVID-19 test experience was reported as “None” by 10 (5.1%) and “Yes” by 186 (94.9%). Perceptions of the importance of the COVID-19 vaccine were “Not important” for 92 (46.9%) and “Important” for 104 (53.1%).

### 3.2. Descriptive Statistics of Influencing Factors and Infection Prevention Behaviors

Descriptive statistics for the influencing factors and infection prevention behaviors are reported in Table 2. Knowledge about COVID-19 had a mean of 12.38 ± 2.00, indicating high knowledge on the topic among adolescents. Meanwhile, attitudes toward COVID-19 and social support had means of 3.25 ± 0.46 and 3.55 ± 0.53, respectively. The government and community responses to COVID-19 were rated 3.49 ± 0.92 and 3.68 ± 0.75 on average, respectively. The mean score for infection prevention behaviors was 3.60 ± 0.53.

### 3.3. Differences in Influencing Factors and Infection Prevention Behaviors Depending on Participants’ General Characteristics

Next, we examined whether the influencing factors and infection prevention behaviors differed according to the participants’ general characteristics. The results are summarized in Table 2. Attitudes toward COVID-19 differed significantly by school class (t = −2.56, *p* = 0.011), life limitations due to COVID-19 (t = −2.40, *p* = 0.017), and the perceived importance of the COVID-19 vaccine (t = −4.13, *p* < 0.001). Social support significantly differed according to subjective physical health status (t = −2.41, *p* = 0.017), mental health status (t = −3.63, *p* < 0.001), life limitations due to COVID-19 (t = −3.20, *p* = 0.002), and perceived importance of the COVID-19 vaccine (t = −2.42, *p* = 0.017).

Perceptions regarding the government’s response to COVID-19 differed significantly by subjective physical health status (t = −3.53, *p* = 0.001), mental health status (t = −2.44, *p* = 0.016), and perceived importance of the COVID-19 vaccine (t = −2.38, *p* = 0.018). Perceptions regarding community response to COVID-19 significantly differed by residential region (t = 2.82, *p* = 0.005), subjective physical health status (t = −2.56, *p* = 0.011), and perceived importance of the COVID-19 vaccine (t = −3.24, *p* = 0.001). Infection prevention behaviors significantly differed according to COVID-19 testing experience (t = −2.10, *p* = 0.037) and the perceived importance of the COVID-19 vaccine (t = −3.67, *p* < 0.001).

### 3.4. Correlations between Different Influencing Factors and Infection Prevention Behaviors

COVID-19 knowledge (r = 0.31, *p* < 0.001) and attitudes toward COVID-19 (r = 0.67, *p* < 0.001), social support (r = 0.46, *p* < 0.001), the government response to COVID-19 (r = 0.19, *p* = 0.008), and the community response to COVID-19 (r = 0.28, *p* < 0.001) were positively correlated with infection prevention behaviors (Table 3).

### 3.5. Factors Influencing Adolescents’ COVID-19 Infection Prevention Behaviors

Hierarchical multiple regression analysis was performed to evaluate the factors affecting adolescents’ COVID-19 infection prevention behaviors. COVID-19 test experience and perceived importance of the COVID-19 vaccine, which showed significant differences in infection prevention behaviors, and gender and life limitations due to COVID-19, which showed a significance of 0.10 or less, were used as control variables. COVID-19 knowledge and attitudes toward COVID-19, social support, the government response to COVID-19, and the community response to COVID-19 were used as independent variables. The results are summarized in Table 4.

Before proceeding with the analysis, we checked for multicollinearity among independent variables using the variance inflation factor (VIF). All VIF values were greater than 1 and less than 10, indicating no multicollinearity issues. The observed values of the dependent variables for all independent variables were normally distributed according to the normal p–p curve.

The explanatory power of Model 1 was 10% (F = 6.21, *p* < 0.001) when only the general characteristic variables were used. The COVID-19 test experience (β = 0.14, *p* = 0.039) and perceived importance of the COVID-19 vaccine (β = 0.25, *p* < 0.001) significantly affected infection prevention behaviors. The explanatory power of Model 2 increased substantially to 50% after adjusting for general characteristics before the test. COVID-19 test experience (β = 0.11, *p* = 0.036), knowledge of COVID-19 (β = 0.11, *p* = 0.048), attitudes toward COVID-19 (β = 0.53, *p* < 0.001), and social support (β = 0.15, *p* = 0.012) significantly affected infection prevention behaviors. Interestingly, the government and community responses to COVID-19 had no significant impact.

## 4. Discussion

This study identified factors influencing COVID-19 infection prevention behaviors among Korean adolescents aged 13–18 years using an ecological model [30]. We found that COVID-19 test experience, intrapersonal factors (knowledge of and attitudes toward COVID-19), and interpersonal factors (social support) had a high explanatory power of 52% for adolescents’ COVID-19 infection prevention behaviors. In contrast, community and government factors had no impact.

Among the intrapersonal factors, the mean knowledge score for COVID-19 among adolescents was 12.38 (out of 16). As no study has been conducted on Korean adolescents’ knowledge of COVID-19, we compared the mean knowledge score to that of a study of college students and found a similar score of 12.46 [34]. In this study, subjects with higher levels of COVID-19 knowledge were more likely to engage in COVID-19 prevention behaviors. This result is consistent with that of a previous study [40]. Next, most adolescents obtained COVID-19 information from various sources, including social media and family, and were more likely to obtain information from friends and parents than from teachers. This has led to gaps in adolescents’ knowledge of COVID-19, preventive measures, and minimally risky activities [40,41]. Because adolescents remain detectable for a significant period and are often asymptomatic or mildly ill, they may act as potential transmitters in the community [19]. Therefore, schools and adolescent-focused organizations should develop content that provides adolescents with reliable, up-to-date information, guidance, and resources to prevent emerging infectious diseases, including COVID-19 [41]. In addition, educational strategies should be considered, such as engaging peers, parents, or caregivers who may influence young people to participate in campaigns under the guidance of an infectious disease specialist to provide up-to-date infectious disease knowledge, information, and training or programs [42].

In this study, better attitudes toward COVID-19 influenced adolescents to engage in better COVID-19 prevention behaviors, which is consistent with previous studies [43,44]. Not many students believe that adolescents are at a higher risk of serious illness due to COVID-19 [40,41]. Nevertheless, adolescents acquired knowledge and information about the new infectious disease from the government, media, and schools and then engaged in COVID-19 infection prevention behaviors such as wearing masks, social distancing, and hand hygiene to protect themselves and others [41,43]. In addition, the results of this study showed that adolescents’ attitudes toward COVID-19 differed depending on their grade, life restrictions caused by COVID-19, and degree of awareness of the importance of the COVID-19 vaccine. This finding is consistent with the results of previous studies [44]. Recognizing how adolescents’ susceptibility to and knowledge of new infectious diseases can play a role in increasing social responsibility, and disease prevention measures can affect attitudes toward new infectious diseases [40,43]. Therefore, it is necessary to develop health education to bring about changes in attitudes toward new infectious diseases such as COVID-19.

Next, as an inter-individual factor, higher levels of social support influenced participants to engage in COVID-19 infection prevention behaviors. This result is similar to previous findings that higher social support increases health prevention behaviors [43]. Since most schools have a high probability of spreading infection from a single infected student to the rest of the classroom and school, as well as to family and community members [12], authorities have attempted to suppress the spread of COVID-19 by closing schools [3]. However, school closures can not only negatively affect adolescents by changing family roles, increasing social isolation, and learning delays [13,14], but can also influence their mental health, including increasing uncertainty and fear due to COVID-19 [14]. Our study also showed that adolescents’ social support differed according to their subjective physical and mental health status, the extent to which COVID-19 limited their lives, and the perceived importance of the COVID-19 vaccine. COVID-19 may become a part of everyday life for years or even permanently [13]. Social support acts as a buffer against adverse, problematic experiences. Social support from the closest family members, friends, and organizations can give individuals confidence, which, in turn, can positively affect their cognitive states and emotions [43]. To focus on family and school environments, promote social connections with friends, and strengthen bonds with peers, it is necessary to help adolescents find what they need; provide appropriate information; and offer emotional, behavioral, and culturally relevant resources [13,14,29]. Based on this, programs should be developed to engage peers, friends, and family to share and learn information and correct misbehavior to help young people understand and change their infection-prevention behaviors [42].

As a community factor, this study found that participants’ perceptions of the government’s response to COVID-19 were better when they had good subjective physical and mental health statuses and perceived the COVID-19 vaccine to be important. Meanwhile, perceptions of the community’s response to COVID-19 were better when they lived in the capital region, had a good subjective physical health status, and perceived the COVID-19 vaccine as important. However, neither the government nor community responses to COVID-19 influenced adolescents’ COVID-19 infection prevention behaviors. This finding differs from those of previous studies [21]. Based on the South Korean experience with the Middle East Respiratory Syndrome (MERS) in 2015, South Korean government agencies and communities communicated with the public from the beginning of the COVID-19 outbreak and engaged them in public health emergency response systems [39]. This may have increased social trust and improved adherence to preventive behaviors [21]. In other words, the sooner the public is engaged, the sooner the outbreak ends [39], which is why timely infection prevention education is important for adolescents who could become potential spreaders. Governments and communities should raise awareness among young people about responding to future emerging infectious diseases such as COVID-19 and ensure that they are able to respond systematically with trust, support, and collaboration [29]. This will require providing short and useful educational media content on subjects considered important by infection experts and disseminating them widely through cyberspace [45].

Similar to previous studies, adolescents were more likely to engage in COVID-19 infection prevention behaviors when they were women, experienced life limitations due to COVID-19, had COVID-19 test experience, and perceived the COVID-19 vaccine as important, similar to previous studies [23,46]. Women were more likely to engage in handwashing as a preventive measure against COVID-19; this may be due to psychological characteristics, such as sensitivity to the risk they face, and cultural characteristics, such as the norms and practices required by society [23,44]. The more anxiety and fear individuals experience during the early stages of the COVID-19 outbreak, the more likely they are to engage in COVID-19 infection prevention behaviors [23]. It is believed that this is because they actively engaged in infection prevention behaviors to relieve themselves of the inconvenience caused by restrictions and difficulties in their daily lives as COVID-19 progressed [47]. Thus, positive—rather than negative—emotions should be promoted [5] so that adolescents can take preventive actions against not only COVID-19 but also other emerging infectious diseases that may arise in the future. To this end, health education programs tailored to the characteristics and preferences of young people should be developed under the supervision of infectious disease experts and delivered through virtual networks with appropriate content [48].

Furthermore, by exploring the factors associated with adolescents’ COVID-19 prevention behaviors based on an ecological model, we found that adolescents had better behaviors to prevent COVID-19 when they had higher levels of knowledge and attitudes toward COVID-19 and positive social support, which is similar to previous studies [43,45]. The threat of COVID-19 continues. While COVID-19 infection rates have decreased, it is more likely to become an endemic disease, similar to influenza or the common cold [48]. However, COVID-19 infections in adolescents show less typical or asymptomatic clinical manifestations than those in adults. Thus, it is necessary to focus on COVID-19 prevention behaviors among adolescents [26]. In addition, parents, schools, and others may face increased practical limitations and stress due to social and economic challenges, even after the outbreak of COVID-19 [49]. Policy communication and supportive efforts are needed to help the government identify and address the social factors and other barriers that prevent adolescents from adopting effective infection prevention behaviors.

## 5. Strengths and Limitations

This study contributes to explaining the significance of individual, interpersonal, and community factors through a multilateral approach based on ecological models for identifying the determinants of COVID-19 prevention behaviors among adolescents.

However, this study has a few limitations. We aimed to maximize the generalizability by selecting three cities distributed across the country. However, as our research primarily focused on adolescents from specific regions, caution should be exercised when generalizing our results. Second, while online surveys offer numerous advantages compared to traditional methods, they may introduce selection bias by excluding adolescents without internet access, potentially affecting the reliability of participant responses.

## 6. Conclusions

This study explored the factors influencing COVID-19 infection prevention behaviors among adolescents in Korea. We found that the factors associated with adolescents’ COVID-19 infection prevention behaviors were COVID-19 test experience, knowledge of COVID-19, attitudes toward COVID-19, and social support, which had an explanatory power of approximately 52%. Based on these results, in order to promote the prevention behavior of new infectious diseases such as COVID-19 in adolescents, education and social support programs that provide youth with knowledge and information related to infectious diseases are necessary. To effectively promote prevention behaviors among adolescents, a multifaceted public health education strategy should be devised, rooted in an ecological model that emphasizes the significance of individual, interpersonal, and community factors.

## Figures and Tables

**Figure 1 healthcare-11-02779-f001:**
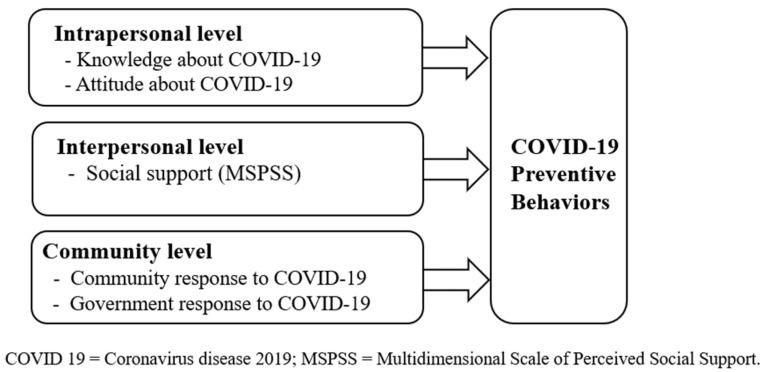
Conceptual model for predicting COVID-19 preventive behaviors.

**Table 1 healthcare-11-02779-t001:** General characteristics of the participants (*N* = 196).

Characteristics	Categories	*n* (%)
Gender	Male	69 (35.2)
	Female	127 (64.8)
School classes	Middle school	98 (50.0)
	High school	98 (50.0)
Residential regions	Capital region	108 (55.1)
	Non-capital region	88 (44.9)
Subjective physical health status	Below average	87 (44.4)
	Good	109 (55.6)
Subjective mental health status	Below average	91 (46.4)
	Good	105 (53.6)
Daily life limitations due to COVID-19	None	111 (56.6)
	Yes	85 (43.4)
COVID-19 test experience	None	10 (5.1)
	Yes	186 (94.9)
Perceived importance of the COVID-19 vaccine	Not important	92 (46.9)
	Important	104 (53.1)

**Table 2 healthcare-11-02779-t002:** Differences among influencing factors and infection prevention behaviors according to general characteristics (*N* = 196).

Characteristics	Categories	Knowledge about COVID-19	Attitudes toward COVID-19	Social Support	Government Response to COVID-19	Community Response to COVID-19	Infection Prevention Behaviors
Gender	Male	12.22 ± 2.20	3.18 ± 0.60	3.52 ± 0.57	3.65 ± 0.94	3.75 ± 0.82	4.50 ± 0.69
	Female	12.47 ± 1.88	3.28 ± 0.35	3.57 ± 0.50	3.40 ± 0.89	3.64 ± 0.72	4.65 ± 0.41
	t/F(*p*)	−0.85 (0.394)	−1.27 (0.207)	−0.64 (0.523)	1.83 (0.069)	0.92 (0.360)	−1.66 (0.100)
School classes	Middle school	12.20 ± 2.24	3.16 ± 0.53	3.51 ± 0.58	3.58 ± 0.88	3.70 ± 0.76	4.55 ± 0.58
	High school	12.56 ± 1.71	3.33 ± 0.36	3.59 ± 0.46	3.40 ± 0.95	3.66 ± 0.75	4.65 ± 0.47
	t/F(*p*)	−1.25 (0.211)	−2.56 (0.011)	−1.04 (0.300)	1.34 (0.181)	0.38 (0.706)	−1.32 (0.187)
Residential regions	Capital region	12.53 ± 1.81	3.25 ± 0.50	3.59 ± 0.51	3.53 ± 0.93	3.82 ± 0.71	4.57 ± 0.59
	Non-capital region	12.20 ± 2.20	3.24 ± 0.40	3.49 ± 0.55	3.44 ± 0.90	3.52 ± 0.77	4.63 ± 0.45
	t/F(*p*)	1.13 (0.260)	0.20 (0.846)	1.32 (0.187)	0.71 (0.480)	2.82 (0.005)	−0.78 (0.435)
Subjective physical health status	Below average	12.52 ± 1.70	3.22 ± 0.43	3.45 ± 0.59	3.24 ± 0.93	3.53 ± 0.74	4.55 ± 0.54
	Good	12.28 ± 2.20	3.27 ± 0.48	3.63 ± 0.45	3.69 ± 0.86	3.80 ± 0.74	4.64 ± 0.52
	t/F(*p*)	0.84 (0.400)	−0.63 (0.529)	−2.41 (0.017)	−3.53 (0.001)	−2.56 (0.011)	−1.12 (0.265)
Subjective mental health status	Below average	12.18 ± 2.31	3.21 ± 0.46	3.40 ± 0.61	3.32 ± 0.86	3.59 ± 0.73	4.57 ± 0.55
	Good	12.56 ± 1.67	3.28 ± 0.45	3.68 ± 0.40	3.63 ± 0.94	3.76 ± 0.77	4.63 ± 0.52
	t/F(*p*)	−1.32 (0.187)	−1.11 (0.268)	−3.63 (<0.001)	−2.44 (0.016)	−1.64 (0.102)	−0.79 (0.433)
Daily life limitations due to COVID-19	None	12.16 ± 2.17	3.18 ± 0.50	3.45 ± 0.58	3.51 ± 0.95	3.75 ± 0.72	4.54 ± 0.55
	Yes	12.67 ± 1.71	3.33 ± 0.38	3.68 ± 0.41	3.46 ± 0.88	3.59 ± 0.80	4.67 ± 0.49
	t/F(*p*)	−1.78 (0.077)	−2.40 (0.017)	−3.20 (0.002)	0.35 (0.723)	1.54 (0.126)	−1.66 (0.098)
COVID-19 test experience	None	11.70 ± 3.27	3.17 ± 0.36	3.37 ± 0.59	3.40 ± 0.60	3.59 ± 0.52	4.26 ± 0.44
	Yes	12.42 ± 1.91	3.25 ± 0.46	3.56 ± 0.52	3.49 ± 0.93	3.69 ± 0.77	4.62 ± 0.53
	t/F(*p*)	−1.11 (0.268)	−0.52 (0.601)	−1.12 (0.264)	−0.31 (0.757)	−0.40 (0.688)	−2.10 (0.037)
Perceived importance of the COVID-19 vaccine	Not important	12.30 ± 1.90	3.11 ± 0.52	3.45 ± 0.59	3.32 ± 0.94	3.50 ± 0.74	4.45 ± 0.62
	Important	12.45 ± 2.08	3.37 ± 0.35	3.63 ± 0.45	3.63 ± 0.87	3.84 ± 0.74	4.73 ± 0.39
	t/F(*p*)	−0.52 (0.607)	−4.13 (< 0.001)	−2.42 (0.017)	−2.38 (0.018)	−3.24 (0.001)	−3.67 (< 0.001)

**Table 3 healthcare-11-02779-t003:** Correlations between major variables (*N* = 196).

Variables	M ± SD	Range	1	2	3	4	5	6
1. Knowledge about COVID-19	12.38 ± 2.00	0–16	1					
2. Attitudes toward COVID-19	3.25 ± 0.46	1–5	0.24 (0.001)	1				
3. Social support	3.55 ± 0.53	1–5	0.32 (<0.001)	0.44 (<0.001)	1			
4. Government response to COVID-19	3.49 ± 0.92	1–5	0.07 (0.332)	0.10 (0.167)	0.14 (0.047)	1		
5. Community response to COVID-19	3.68 ± 0.75	1–5	0.15 (0.036)	0.21 (0.003)	0.27 (<0.001)	0.63 (<0.001)	1	
6. Infection prevention behaviors	4.60 ± 0.53	1–5	0.31 (<0.001)	0.67 (<0.001)	0.46 (<0.001)	0.19 (0.008)	0.28 (<0.001)	1

**Table 4 healthcare-11-02779-t004:** Factors influencing COVID-19 preventive behaviors among adolescents (*N* = 196).

Variables	Model 1			Model 2		
	B	β	t (p)	B	β	t (p)
Gender	0.13	0.12	1.71 (0.089)	0.09	0.08	1.57 (0.117)
Daily life limitations due to COVID-19	0.11	0.10	1.47 (0.144)	−0.02	−0.02	−0.41 (0.679)
COVID-19 test experience	0.34	0.14	2.08 (0.039)	0.26	0.11	2.11 (0.036)
Perceived importance of the COVID-19 vaccine	0.27	0.25	3.71 (0.000)	0.05	0.05	0.93 (0.356)
Knowledge about COVID-19				0.03	0.11	1.99 (0.048)
Attitudes toward COVID-19				0.62	0.53	9.04 (0.000)
Social support				0.15	0.15	2.54 (0.012)
Government response to COVID-19				0.04	0.08	1.16 (0.247)
Community response to COVID-19				0.03	0.04	0.65 (0.519)
R^2^	0.12			0.52		
adjusted R^2^	0.10			0.50		
F(*p*)	6.21 (<0.001)			22.64 (<0.001)		

Reference of dummy variables: Gender = Male; Life limitations due to COVID-19 = None; COVID-19 test experience = None; Perceived importance of the COVID-19 vaccine = Not important.

## Data Availability

The data presented in this study are available upon request from the corresponding author. The data are not publicly available to protect the privacy of the research subjects.

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
