# Peer review of "Factors Related to the COVID-19 Prevention Behaviors of Adolescents: Focusing on Six Middle and High Schools in Korea"

_healthcare, 2023, doi:10.3390/healthcare11202779_

Round 1
Reviewer 1 Report (Previous Reviewer 1)
In Strengths and limitations ... ecological model to identify. ecological?? Conclusions is long.Author Response
We have attached the reviewer's response

Reviewer 2 Report (Previous Reviewer 3)
The authors corrected the manuscript according to the reviewer's comments. Let me repeat my previous opinion: the thesis does not contain scientific innovation, surprising results, it is a relatively simple survey but it properly directs attention to the insufficient health literacy of adolescents, which may claim lives in public health emergencies.
Author Response
We have attached the reviewer's response

Reviewer 3 Report (Previous Reviewer 4)
The study titled "Factors Related to COVID-19 Prevention Behaviors of Adolescents in Korea" aims to investigate the determinants of COVID-19 preventive behaviors among adolescents. The study has several notable weaknesses, including a small cross-sectional sample size that is not representative of a broader geographical region and the use of an online survey. Furthermore, the topic's novelty is questionable, given the abundance of existing research on COVID-19 prevention behaviors.
Adolescents in Korea represent a diverse population, and this study's limited sample may not adequately capture the variation in attitudes and behaviors across different regions, socioeconomic backgrounds, and demographic characteristics. COVID-19 prevention behaviors can vary widely based on cultural, economic, and regional factors. Therefore, the study's findings may not be applicable to all adolescents in Korea, let alone adolescents in other countries.
Online surveys are prone to selection bias as it may exclude adolescents who do not have internet access or those who are less likely to engage in online activities. Additionally, online surveys can be limited in the depth of information they collect. The COVID19 also led to an overwhelming number of online surveys.
By the time the study was conducted (between December 2022 and May 2023), the global scientific community had accumulated a substantial body of research on COVID-19 prevention behaviors. Given the fast-evolving nature of the pandemic, research priorities shifted towards more advanced topics such as vaccine acceptance, vaccine hesitancy, and long-term effects of the pandemic. Consequently, the study may not contribute significantly to the existing literature, and its findings may have limited practical implications beyond reiterating well-established factors.
Author Response
We have attached the reviewer's response

Reviewer 4 Report (New Reviewer)
Dear authors, thank you for let me reviewing your manuscript titled “Factors related to COVID-19 prevention behaviors of adolescents in Korea”.
I see as a cross-sectional study from a convenience sample from six Sur-Corean Schools.
I appreciatte the author effort but, unfortunately, there are some flaws that, in my opinion, limit its publication.
The main one is the sample size calculation, considering the Sur-Corean adolescents populations and the inclusion criteria setled by authors. Accordingly to this, the results should be limited to this six schools, and from my point of view, could be useful to a local report.
Furthermore, the conclusions (lines 19 to 28 in the abstracts) looks like authors opinions instead of evidence from the results.
On other hand, I disagree with some authors statements , like that in line 56-57 “This indicates that COVID-19 has had more severe effects on adolescents than on adults…” Or in line 304 “… adolescents are at a higher risk of serious illness due to COVID-19
I suggest authors considering the COVID-19 adolescents mortality /morbility versus adults one. Or the long COVID-19 disease (post-COVID) prevalence on adolescents compare to adults.
Author Response
We have attached the reviewer's response

Round 2
Reviewer 3 Report (Previous Reviewer 4)
After reviewing the author's responses to my previous comments, I appreciate their efforts to address some of the concerns raised. However, I still believe that the paper lacks the required level of novelty given the abundance of existing research on COVID-19 prevention behaviors. Furthermore, the use of an online survey and the limitations associated with sample representativeness in terms of geographical and demographic diversity remain concerns. Given these factors, I maintain my decision to reject the manuscript, and I suggest that the authors consider submitting their work to a more localized journal where the specific context of Korean adolescents may be of greater relevance.
Author Response
There may be similar papers overseas, but papers targeting the Korean population are rare. Due to the high incidence of COVID-19 in Korea in 2022, it was difficult to conduct a survey of youth due to concerns about infection. We worked hard to collect data at a time when access to youth was difficult. I would like to have the opportunity to inform foreign readers of research related to the factors influencing infection prevention behavior among some Korean adolescents.
As commented by the reviewer, there is an abundance of existing research worldwide concerning COVID-19 prevention behaviors. However, the majority of research results, which primarily focus on their respective countries rather than Korea, may not directly translate to the context of Korean adolescents due to different cultural and economic factors. In the realm of studies involving Korean adolescents, most research has predominantly applied the Health Belief Model and Theory of Reasoned Action. The significance of this study lies in its application of the Ecological model to adolescents, which is expected to provide valuable insights for the development of practical COVID-19 prevention behavior guidelines tailored to Korean adolescents.
s pointed out by the reviewer, COVID-19 prevention behaviors can exhibit significant variations influenced by cultural, economic, and regional factors. In contrast to large countries like USA, the school system in Korea remains relatively consistent across regions. However, in order to account for even the minor variances, we have taken steps to ensure sample diversity by collecting data from three major cities located in different regions of Korea: one in Seoul, another in the southwestern part, and the third in the southeastern part.
Reviewer 4 Report (New Reviewer)
Dear authors I applaud your effort to improve the manuscript, changing the title and modifying some paragraphs from methodology, limitations and conclusions.
Author Response
Thank you for your feedback on the manuscript.

This manuscript is a resubmission of an earlier submission. The following is a list of the peer review reports and author responses from that submission.
Round 1
Reviewer 1 Report
In tittle: if possible If possible, change Affecting to related to...
At the beginning of the abstract, add 1 line about the importance of preventive behavior from Covid.
In abstract: In the method section, the method of analysis should be stated.
Line 62 needs reference.
At the end of the introduction; the aim should be clearly stated.
Study Design not clear, Study Design for example: cross section study…
This study was a descriptive study to identify factors related to Korean adolescents' 96 infection prevention behaviors based on McLeroy et al.'s ecological model [26]. This is ambiguous.
This is a cross section study not ecological model.
Sample size is low and more sample size is needed.
Sampling method should be stated.
In table 1, M±SD not clear.
In table 2 what is t/F?
If possible, Table 2 should be summarized.
Backward regression??
In the discussion of all the sentences, it needs a reference.
doi: 10.3389/fpubh.2020.585302. eCollection 2020. And doi: 10.1007/s12029-021-00679-x. Epub 2021 Aug 18. May be useful for improving the discussion.
Reviewer 2 Report
The manuscript presents a descriptive study exploring which factors influence adolescents' COVID-19 prevention behaviors. Due the theme (COVID-19) the references are very actuals.
This study can contribute to depth knowledge about this topic. There is a need for a more accurate explanation of the possible contribution of the research in terms of education strategies (school compares with social media and family) and innovative approaches to learning in these ages and the research limitation.
Congratulations, there is an important paper for advanced knowledge in this area.
Reviewer 3 Report
The South Korean study seems to support the popular belief that young people's COVID-19 prevention behaviour is primarily influenced by social media and information obtained from peer groups, and not by organized health promotion, government influence or school education. This can be experienced in many countries, and this is even the case in South Korea, which is performing well in the management of the COVID pandemic. The topic is not original, but it causes a serious and measurable loss in lives if health policy and society do not deal with it. The findings do not contain much news for health promoters and advocates, but this problem is worth discussing.
As I assume the authors' main question is where the health knowledge related to COVID and its preventability comes from. To this they give a disturbing answer, namely that it is not from a government or organized educational source.
The parts of the introduction, the methodology and the results are adequate, although perhaps it would be advisable to include the questionnaire in its entirety as part of the article. It would be worthwhile to classify the answers based on the income and the social situation of the parents, if there is data for this. The methodology makes up a disproportionately large part of the draft, it is undoubtedly difficult to understand, bar and pie charts would be nice, as well as the attachment of the questionnaire. The whole chapter can be shortened.
The abstract and the conclusion do not overlap, the latter contains ambiguous statements (e.g. the COVID testing experience appearing in the latter, which is poorly explained). The conclusions are moderate, although among the recommendations, I personally consider it necessary to include health science as a subject in primary and secondary school curricula, with due attention to the often neglected infectious diseases.
Tables are complicated, presenting too many variables instead of focusing on the main question.
It is interesting that most of the references are domestic publications, more international and WHO references would be useful.
Reviewer 4 Report
Thank you for the opportunity to review the manuscript titled “Factors Affecting COVID-19 prevention behaviors of adolescents in Korea”.
I reviewed this work to determine suitability for publication based on a combination of factors, including whether the topic is well suited to the aims and scope of the journal, methodological considerations, and whether the findings make a sufficient contribution to the existing literature. Unfortunately, I believe the manuscript does not meet these requirements.
The manuscript does not clarify how the study's objectives or findings contribute to the current body of knowledge on COVID-19 prevention behaviors. It is crucial to ensure that the study's focus and conclusions align with the latest understanding of the topic.
With the methods used (e.g., online survey) and the small sample size recruited, not enough is done to demonstrate that the findings here generalize beyond current study. I do not see unique contributions, such as novel methodologies, novel sample populations, or addressing novel research gaps.
All the aforementioned issues make the conclusions very limited and difficult to be utilized.
Thanks,
none
Reviewer 5 Report
1. The significance of studying COVID-19 in adolescents is not clear: This study focuses on adolescents and analyzes 196 valid samples of individuals aged 13 to 18 through online surveys. As the actual physical and psychological impacts of COVID-19 on adolescents lack direct literature evidence, and COVID-19 vaccines for this age group were developed later, it is crucial to clearly define the scope of the study concerning the differences in COVID-19 prevention and treatment between adolescents and adults. This will highlight the importance of this research and demonstrate its value. For instance, does this study consider the significance of studying adolescents in relation to the different "transmission settings (e.g., classrooms)" among Korean middle school students? However, the paper also points out, "...Since adolescents remain detectable of viruses for a significant period of time and are often asymptomatic or mildly ill, they may act as potential transmitters in the community." (Lines 304-305), indicating that adolescents and adults differ in COVID-19 prevention and treatment. Nonetheless, the paper lacks a comprehensive interpretation of the uniqueness and value of studying COVID-19 prevention in adolescents. For example, although the group lifestyle of Korean adolescents (Lines 44-63) may make them more susceptible to COVID-19 transmission, it does not necessarily imply that adolescents will experience more severe symptoms. Adolescents' information reception and dissemination are naturally different. The study should provide a more detailed differentiation regarding the uniqueness and association of COVID-19 among the study subjects "adolescents." Otherwise, it may not adequately demonstrate the implications and insights that the research on Korean adolescents' gatherings can provide to different countries and societies, as there have been severe COVID-19 transmissions in the past related to "religious gatherings" in Korea. Therefore, the paper fails to present what potential lessons and references the study on Korean adolescents' gatherings can offer.
2. Issues with the representativeness of the research sample: Although this paper follows Cohen's sampling, it may still face criticism for having a small sample size and lacking representativeness. Since this study deals with an infectious disease issue concerning the concept of "adolescent group living conditions," it is necessary to provide detailed explanations of various characteristics of Korean adolescent groups. The discussion of the representativeness should also be more comprehensive and elaborate.
3. This is a descriptive study, but from the "Results" (Line 214-) to the "Discussion" (Line 288-), the paper fails to clearly understand the relationships between "variables and variables" and the possible existence of "mediating variables" between them. For example, "attitudes toward COVID-19, life limitations due to COVID-19, and perceived importance of the COVID-19 vaccine, social support significantly differed by subjective physical health status, mental health status, life limitations due to COVID-19, and perceived importance of the COVID-19 vaccine." Why do these variables have correlations? Are there any mediating variables between them? These aspects need thorough discussion and classification. The paper lacks precision in dealing with similar concepts.